# Association between Dietary Diversity Changes and Cognitive Impairment among Older People: Findings from a Nationwide Cohort Study

**DOI:** 10.3390/nu14061251

**Published:** 2022-03-16

**Authors:** Dan Liu, Wen-Ting Zhang, Jia-Hui Wang, Dong Shen, Pei-Dong Zhang, Zhi-Hao Li, Pei-Liang Chen, Xi-Ru Zhang, Qing-Mei Huang, Wen-Fang Zhong, Xiao-Ming Shi, Chen Mao

**Affiliations:** 1Department of Epidemiology, School of Public Health, Southern Medical University, Guangzhou 510515, China; liudan0717@smu.edu.cn (D.L.); mos274@163.com (W.-T.Z.); wjh385691886@126.com (J.-H.W.); shendong@smu.edu.cn (D.S.); zpd911@gmail.com (P.-D.Z.); zhihaoli2013@163.com (Z.-H.L.); peiliangchen51@hotmail.com (P.-L.C.); xiruzhang0130@163.com (X.-R.Z.); 96qingmei@i.smu.edu.cn (Q.-M.H.); zhongwf0613@hotmail.com (W.-F.Z.); 2Department of Public Health and Preventive Medicine, School of Medicine, Jinan University, Guangzhou 510632, China; 3National Institute of Environmental Health, Chinese Center for Disease Control and Prevention, Beijing 100021, China; 4Department of Laboratory Medicine, Microbiome Medicine Center, Zhujiang Hospital, Southern Medical University, Guangzhou 510280, China

**Keywords:** cognitive impairment, dietary diversity changes, older adults, cohort study

## Abstract

Background: Little is known about the role of dietary diversity changes in affecting cognitive function among older people. Therefore, we aimed to evaluate the associations between dietary diversity scores (DDS) changes with cognitive impairment among older adults in a large prospective cohort. Methods: Cognitive function was assessed using the Mini-Mental State Examination questionnaire at baseline and follow-up. A total of 9726 participants without Parkinson’s disease, dementia, or cognitive impairment were enrolled at baseline. Nine food groups were collected using simplified FFQ at baseline and follow-up surveys. Then nine food groups change patterns and DDS change patterns (overall, plant-based and animal-based) were assessed. The associations of above DDS changes patterns with subsequent cognitive impairment were evaluated. A multivariable-adjusted Cox proportional hazards model was used to estimate HRs and 95%CIs. Results: We documented 2805 cognitive impairments during 52,325 person-years of follow-up. Compared to high-to-high overall DDS change patterns, the multivariable adjusted HRs (95%CI) for high-to-medium, medium-to-medium, medium-to-low, low-to-medium and low-to-low DDS change patterns were 1.33 (1.12–1.57), 1.11 (0.94–1.32), 1.61 (1.39–1.86), 2.00 (1.66–2.40), 2.30 (1.90–2.78) and 2.80 (2.23–3.53), respectively. Compared with participants with stable DDS change pattern, those who in large improvement of DDS had a 13% lower risk of cognitive impairment (HRs, 0.87; 95%CI: 0.78–0.98). The associations of plant-based DDS, animal-based DDS, or nine food groups DDS change patterns with cognitive impairment were in a similar direction to the main result. Conclusions: Protective associations between maintaining high DDS and a reduced risk of cognitive impairment were observed. In contrast, lowering or maintaining a lower DDS increases the risk of cognitive impairment.

## 1. Introduction

Rapidly aging population growth poses a significant challenge to the aging society to maintain the cognitive vitality of the elderly. As a major clinical and public health concern, cognitive decline potentially threatens the quality of life of the elderly and their families with no significantly effective treatment currently, and the prevention of cognitive impairment in older adults has therefore become increasingly essential [1]. Identifying possible modifiable protective and risk factors may allow for early interventions to delay the onset of cognitive impairment or dementia [2].

Diet is a key modifiable factor in age-related diseases [3,4,5]. Accumulating epidemiological evidence suggests that diet plays an important role in inhibiting the onset of age-related diseases [6,7,8,9,10]. In this light, there has been increased interest shown by investigators worldwide in dietary patterns [11,12,13,14,15,16,17,18], including the Healthful Plant-Based Diet Index (HPDI) [11,12], the Healthy Eating Index–2015 (HEI-2015) [13], the Alternate Healthy Eating Index-2010 (AHEI-2010) [14] and the Alternate Mediterranean Diet (AMED) [15], etc. Accumulated evidence supports an association between high-quality diet and cognitive impairment [14,19,20,21,22,23,24,25,26], and indicates that higher adherence to the high-quality diet to be associated with better cognition in older people [27,28,29].

Dietary diversity (DD) takes into account the fact that individuals do not consume isolated nutrients, single meals or certain food groups but rather eat meals consisting of a variety of foods with multiple nutrients. DD is now recognized globally as a vital element of high-quality diet, for a high diet quality is a combination of multiple components that act synergistically [30]. However, the previous studies mostly measured the baseline DD as the relevant exposure but ignored the potential diet fluctuations over time during follow-up, which may introduce some measurement error on the one hand. On the other hand, it remains unclear how the dietary diversity score (DDS) changes during follow-up may alter cognitive function.

Therefore, this study aims to examine DDS change and the risk of cognitive impairment among the Chinese elderly aged 65 years or older from 1998 based on the Chinese Longitudinal Healthy Longevity Survey (CLHLS), an ongoing prospective cohort study in China. Further, the associations between DDS composed of nine food groups and cognitive impairment were accessed to identify key food groups that could be targeted to help maintain better cognitive function.

## 2. Methods

### 2.1. Study Setting

The CLHLS is a prospective cohort study of the Chinese population aged 65 years or over. Participants were enrolled in waves of 1998, 2000, 2002, 2005 and 2008-09 and have been followed up ever since. Participants were recruited in 806 cities and counties randomly selected from 23 provinces across the country by using a multistage stratified sampling method, covering approximately 85% of China’s population. For each sampled centenarian in the baseline survey, one nearby nonagenarian (90–99 years), octogenarian (80–89 years) and younger elderly (65–79 years) of predefined sex to match with the centenarian were interviewed. Based on the randomly assigned centenarians’ code numbers, the predefined sex was determined to obtain comparable numbers of women and men for each age group. Detailed descriptions of the study design and data quality assessment have been provided in previous publications [31,32]. At each follow-up period, information on diet, lifestyle and medical history was updated through face-to-face interviews using structured questionnaires.

In this study, we used five waves of CLHLS data collected in 1998, 2000, 2002, 2005 and 2008–2009. Inclusion criteria were as follows: (1) age 65 or over; (2) with no Parkinson’s disease, dementia, or cognitive impairment at baseline; (3) successfully completed two follow-up surveys and obtain available diet information at baseline and the first follow-up. Among 42,147 participants who were recruited in CLHLS survey from 1998–2009, we excluded 20,587 participants who died (*n* = 14,105) or were lost to follow-up (*n* = 6482) at the first follow-up survey. Among the 21,560 participants available at the first follow-up, we also excluded participants who died (*n* = 6926) or were lost to follow-up (*n* = 2835) before the second follow-up survey. Of the 11,799 participants who were available at the second follow-up survey, we excluded those with Parkinson’s disease (*n* = 49), dementia (*n* = 35), or cognitive impairment (*n* = 1390) at baseline or lacked dietary data at baseline or first follow-up survey (*n* = 450), or who were less than 65 years of age (*n* = 149). A total of 9726 participants were included in this study (Appendix A).

### 2.2. Assessment of DDS

In face-to-face interviews, dietary information was collected using a validated simplified FFQ without quantity in Chinese [33,34], and the information included nine food groups as follows: fresh vegetables, fresh fruit, tea, garlic, food made from beans, meat, fish, eggs and preserved vegetables. Participants were asked how often these foods were consumed, and the dietary information was recorded as frequent (≥5 times/week), occasional (1–4 times/week) or rare (<1/week). Considering the frequency of food consumption, we counted the number of food groups to measure the DDS, the nine food groups were given a score of 0 (rare), 1 (occasional) and 2 (frequent) without considering a minimum intake for the food groups [32,35]. Thus, the score of overall DDS, plant-based DDS (including fresh vegetables, preserved vegetables, fresh fruit, tea, garlic and food made from beans) and animal-based DDS (including meat, fish and eggs) was calculated based on a scale of 0–18, 0–12, and 0–6, respectively.

### 2.3. Assessment of DDS Change Patterns

We calculated the overall DDS at baseline and first follow-up survey from nine food groups and categorized them into three groups: high (13–18 score), medium (7–12 score), and low (0–6 score). Similarly, the plant-based DDS was categorized into high (9–12 score), medium (5–8 score), and low (0–4 score) groups, while, the animal-based DDS was categorized into high (5–6 score) medium (3–4 score) and low (0–2 score) groups. Then, nine relative DDS change patterns were created as follows: high-to-high, high-to-medium, high-to-low, medium-to-high, medium-to-medium, medium-to-low, low-to-high, low-to-medium, and low-to-low.

Moreover, the absolute change scores of the DDSs were calculated using DDS at baseline and the first follow-up, including extreme decline (score ≤ −5), moderate decline (score of −4 to −2), stable (score of −1 to 1), moderate improvement (score of 2 to 4), and large improvement (score ≥ 5). The baseline characteristics according to absolute DDS change patterns are summarized in Appendix A.

### 2.4. Ascertainment of Cognitive Impairment

The primary outcome of the present analysis was cognitive impairment. Cognitive function at baseline and follow-up surveys was evaluated using the Chinese version of the Mini-Mental State Examination (MMSE), and the scale includes the following six areas: orientation, registration, attention and calculation, language, memory, and visual construction skills [36]. A total of 30 items were scored, A score of zero was given for incorrect and unknown answers, and one point was given for correct answers, thus the maximum score 30 points for each participant. We defined cognitive impairment with the education-base cutoff value: 20 points for illiterate, 23 points for participants with 1–6 years of education, 27 points for those with more than 6 years of education [37].

### 2.5. Ascertainment of Covariates

Covariate information on sociodemographic, lifestyle, and comorbidities were obtained from the structured questionnaire for the baseline survey: (1) sociodemographic factor, including age, sex, residence, education level, occupation, source of income, current marital status, and living pattern; (2) lifestyle habits, including smoking status, alcohol drinking, BMI, physical activity, use of artificial dentures; (3) comorbidities, including self-reported chronic diseases: hypertension, diabetes, heart diseases, cerebrovascular diseases, respiratory diseases, digestive system diseases, activities of daily living (ADL) disabled, cancer, eye diseases, and arthritis. We calculated BMI as the weight in kilograms (kg) divided by the square of the height in meters (m^2^), and categorized it into <24.0 and ≥24.0 kg/m^2^. The Katz Index of Independence was applied to assess ADL disabled, respondents who needed assistance in performing one of the following ADLs were considered as ADL disability: eating, toileting, bathing, dressing, indoor activities and continence [38].

### 2.6. Statistical Analysis

Baseline characteristics are presented as the number (percentage) for categorical variables and the mean (with standard deviation, SD) for continuous variables. “Person-years” are calculated from the time of the baseline survey of participants to the earliest of the following events (first occurrence of cognitive impairment, death, lost to follow-up or time of the last survey). Less than 5% of the data for the study covariates were missing (Appendix A), and we used multiple imputation by chained equations (MICE) to impute any missing covariate values with 10 datasets.

The relationship between DDS change patterns and cognitive impairment were explored using Cox proportional hazard models with time. We examined the proportional hazards assumption by creating a cross product of follow-up time and DDS change patterns. For each covariate, the Cox proportional hazards assumption was evaluated with Kaplan–Meier curves, and no major violations were observed.

Three sets of models were used. Model 1 was adjusted for baseline age (continuous) and sex (male or female); In model 2, we further adjusted for residence (urban or rural), education level (no schooling, ≤6 years or >6 years), occupation (worker, farmer or others) and source of income (pension or others). In model 3, additional variables were controlled for current marital status (married or not married), living pattern (living with family members, alone, or in a nursing home), tobacco smoking (current, former or nonsmoker), alcohol drinking (current, former or nondrinker), regular exercise (yes or no), BMI (continuous), use of artificial dentures (yes or no), self-reported hypertension (yes or no), diabetes (yes or no), heart diseases (yes or no), cerebrovascular diseases (yes or no), respiratory diseases (yes or no), digestive system diseases (yes or no), ADL disabled (yes or no), cancer (yes or no), eye diseases (yes or no) and arthritis (yes or no).

The interaction analyses were performed according to baseline age (65–79, ≥80 years), sex (male or female), current marital status (married or not married), tobacco smoking (current and former, or nonsmoker), alcohol drinking (current and former, or non-drinker), regular exercise (yes or no), use of artificial dentures (yes or no) and ADL disabled (yes or no). To test the robustness of our primary findings, we conducted two sensitivity analyses: (1) participants with self-reported prevalent chronic diseases, including hypertension, diabetes, heart diseases, cerebrovascular diseases, respiratory diseases, digestive system diseases and cancer at baseline were removed to minimize the influence of reverse causation; and (2) those who developed cognitive impairment within the fourth year of follow-up were excluded so as to evaluate whether the relationships are consistent over time.

We performed all analyses using SAS software (version 9.4 for Windows, SAS Institute, Inc., Cary, NC, USA). A two-sided *p*-value less than 0.05 was considered statistically significant for statistical tests.

## 3. Results

### 3.1. Baseline Characteristics

Table 1 shows the baseline characteristics of the study participants stratified by nine DDS change patterns (high-to-high, high-to-medium, high-to-low, medium-to-high, medium-to-medium, medium-to-low, low-to-high, low-to-medium, and low-to-low). Of the 9726 participants, 4644 (47.8%) were male, with a mean (SD) age of 80.0 (10.1) years. Of them, 76.0% lived in rural areas, nearly one quarter (24.6%) consumed alcohol, 23.6% were current smokers, and 36.9% reported regular engaging in exercise. The highest number of participants belonged to the medium-to-medium group (4012, 41.3%), while, the high-to-low group had the least (115, 1.2%). Being aged 65–79, being male, being an urban resident, having a higher educational level, having a pension, being married, living with family members, currently smoking, currently drinking, taking regular exercise, using artificial dentures, having higher BMI, having fewer digestive system diseases, without ADL disability and having lower severity of eye diseases were predictors of the high-to-high group.

### 3.2. Association between DDS Change Patterns and Cognitive Impairment

A total of 2805 participants who had normal cognitive function at baseline developed cognitive impairment during 52,325 person-years of follow-up, and the high-to-high group had the lowest incidence rate of cognitive impairment (34.4 per 1000 person-years), while, the low-to-low group had the highest (89.7 per 1000 person-years) (Figure 1 and Appendix A). Figure 1 presents DDS change patterns and their associations with the risk of cognitive impairment. Compared to participants with high-to-high of overall DDS pattern, those who in high-to-medium, medium-to-medium, medium-to-low, low-to-medium and low-to-low DDS pattern groups had a higher cognitive impairment risk, the HRs (95%CI) were 1.33 (1.12–1.57), 1.11 (0.94–1.32), 1.61 (1.39–1.86), 2.00 (1.66–2.40), 2.30 (1.90–2.78) and 2.80 (2.23–3.53) respectively. The association between plant-based DDS or animal-based DDS and cognitive impairment were largely similar to the main results. As for plant-based DDS and animal-based DDS, participants in the high-to-low group had higher cognitive impairment risk compared to the high-to-high group with HRs (95%CI) of 1.49 (1.11–2.01) and 1.30 (1.01–1.66) (Figure 1 and Appendix A).

In Figure 2, we show the association between absolute DDS change groups (extreme decline, moderate decline, stable, moderate improvement and large improvement) and cognitive impairment. Compared with participants with stable DDS change, those who in large improvement of DDS had a 13% lower risk of cognitive impairment (HRs, 0.87; 95%CI: 0.78–0.98), and there was a nonstatistically significant increase in risk for cognitive impairment in other groups (*p* > 0.05). The associations between plant-based or animal-based DDS and cognitive impairment were in a similar direction to the main result (Figure 2 and Appendix A).

Moreover, the associations between cognitive impairment and DDS change patterns in nine foods (including fresh fruit, vegetables, tea, etc.) are presented in Table 2. These results were highly similar to the main results presented herein.

### 3.3. Subgroup and Sensitivity Analyses

We conducted stratified analyses according to potential risk factors (Table 3 and Appendix A). After fully adjusting for the covariates, we observed statistical interactions between DDS change patterns and baseline age (65–79, ≥80), regular exercise (yes or no) and ADL disabled (yes or no) on the risks of cognitive impairment. When stratified by sex, marital status, tobacco smoking, alcohol drinking and use of artificial dentures, the associations were similar to our main results. Sensitivity analyses were robust with no substantial change when we excluded participants who with self-reported prevalent chronic diseases and developed cognitive impairment within the fourth year of follow-up (Appendix A).

## 4. Discussion

In this large population-based cohort study, we found that maintaining a highest DDS was associated with a significantly lowest risk of cognitive impairment. The risk of cognitive impairment was significantly lower among participants who maintained a higher DDS than among those who had consistently lower DDS patterns over time. Those whose DDS improved largely were associated with a decreased risk of cognitive impairment in subsequent years. The associations were independent of cognitive impairment risk factors and other dietary factors.

In accordance with previous studies associating dietary patterns with cognitive impairment, higher diet quality scores measured with the HPDI, HEI-2015, AHEI-2010 and AMED were associated with lower cognitive impairment risk [14,19,20]. However, given that the means age of the participants was as high as 80.0 years and the low education level, we considered that diet quality scores measured above might be difficult for the elderly, and DDS without quantitative measurements might be the most sensible choice. Further, the association between DDS changes and the subsequent cognitive impairment risk is unclear yet. In our study, we found that changes in DDS patterns were associated with subsequent cognitive impairment risk. Large and moderate improvements in DD over time could meaningfully decrease the risk of cognitive impairment, and conversely, worsening DD may increase the risk. This finding may be explained by the fact that DDS can be seen as a proxy and quick indicator of nutrient adequacy, and increasing the variety of food groups in the diet is positively associated with adequate nutritional intake [39,40], maintaining a higher DDS thus is associated with better cognitive function. In contrast, maintaining a lower DDS may mean malnutrition, which will increase the risk of cognitive impairment. However, a review shows that dietary changes are present in most people with dementia, with the progression of dementia, the patient’s ability to obtain adequate nutrition decreases. This could potentially result in reverse causation bias, where changes in dietary may be a consequence of cognitive impairment rather than a cause. Therefore, we exclude those who developed cognitive impairment within the fourth year of follow-up to evaluate whether the relationships are consistent over time, and the sensitivity analyses were robust [41]. Overall, improving DDS or maintaining high DDS is critical for adequate nutritional intake, and thus might reduce the risk of cognitive impairment, this is in accordance with the findings of a multidomain intervention trial in Finnish [29]. These data have important public health implications for the maintenance of normal cognitive function in older adults. The concordance of our observations with previous studies further underscores the importance of maintaining a high-quality diet or improving the DDS.

Our findings are also consistent with the statement of the 2015 Dietary Guidelines Advisory Committee that it is no need to adherence to a single diet plan to achieve healthy eating patterns [42]. Instead, individuals should consume a variety of foods that are healthful as dietary guidelines recommended to fulfill the nutritional needs [43,44,45,46,47]. When we repeated the analysis separately based on plant-based and animal-based DDS changes, and results were also similar to the main results. In addition, maintaining a higher consumption of the nine foods was associated with lower risk of cognitive impairment. Though different in description and composition with DDS mentioned above, we all capture the essential elements of a healthy diet such as fresh fruits, fresh vegetables and fish.

Another interesting finding in our study was the interaction effect of age (65–79 or ≥80), regular exercise (yes or no), and ADL disabled (yes or no) with DDS change patterns from baseline to first follow-up on cognitive impairment. The possible explanation is that the risk of cognitive impairment was increased with age [48]. Further, the elderly in older age groups with poor physiology function in ingestion and absorption, which influence the nutrient intake, might lead to the higher risk of cognitive impairment [49]. Regarding regular exercise, it may mitigate ageing-related cognitive decline [50], and older adults with regular exercise are more likely to consume variety of foods with a high level of concern for health. For the same DDS changes pattern, they will participate in a greater extent when compared with participants without regular exercise, and finally they will get more benefit for their cognitive function. As for ADL disabled, our findings are in agreement with earlier studies showing that older people with ADL disabled are more likely to develop cognitive impairment [51]. This finding suggests that maintaining a higher DDS at early age is important. Based on a higher DDS, taking physical exercise and with a good ability of daily living among older adults should be viewed as a public health intervention to address cognitive impairment benefiting.

The strengths of this study include a prospective design, large sample sizes, high rates of follow-up, repeated assessment of diet and covariates, robust results of sensitivity analyses, and an appropriate approach among older adults to calculate DDS. However, it is worth noting a few limitations of the present study. Firstly, our study evaluated only short-term changes (about 3 years) in DDS, but dietary changes in DDS may have occurred earlier in life or across a longer time-span. Secondly, the collected dietary information lacks quantitative dietary intake, which was not available from the FFQ, thus we were unable to adjust the energy intake model and address whether DDS is associated with total caloric intake. However, a number of key determinants of energy intake were taken into account, such as age, sex, BMI, comorbidities and physical activity [47,52]. Thirdly, the food groups we access do not include nuts and milk, as most older people are rural people who cannot afford them. Finally, although we adjusted for as many potential confounders as possible, such as sociodemographic, lifestyle and comorbidities, residual and unmeasured confounding could not be completely ruled out in this observational study.

## 5. Conclusions

In conclusion, among Chinese older adults, we observed the protective associations between maintaining higher DDS and a reduced risk of cognitive impairment. In contrast, lowering or maintaining a lower DDS increases the risk of cognitive impairment.

## Figures and Tables

**Figure 1 nutrients-14-01251-f001:**
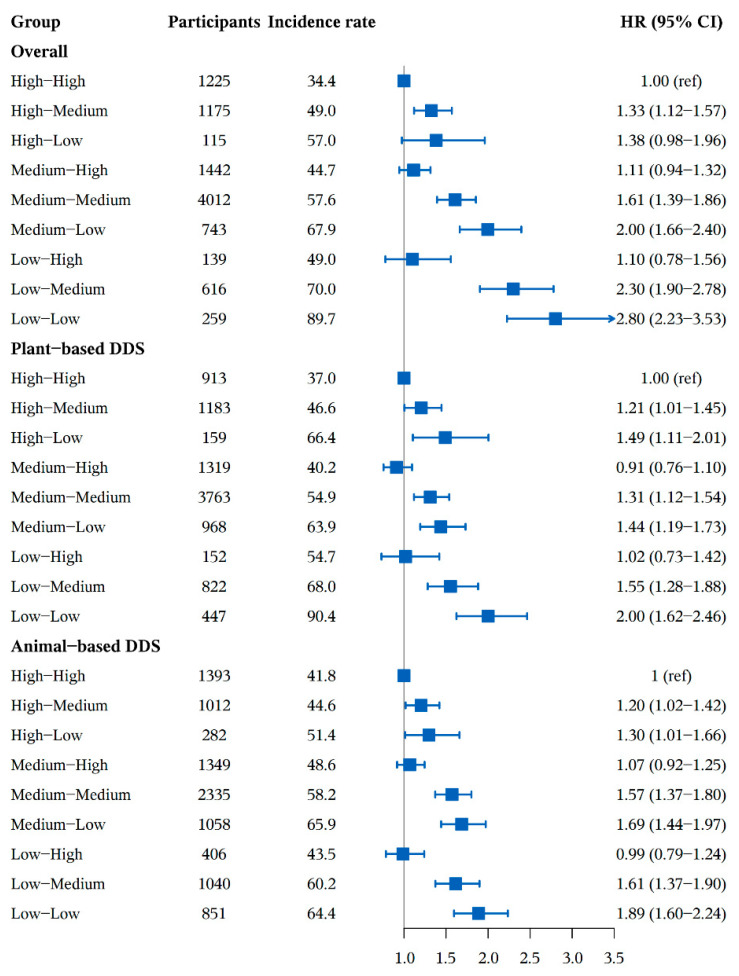
The association between relative DDS change patterns and cognitive impairment. Adjusted for age (continuous), sex, residence, educational level, occupation, source of income, current marital status, living pattern, tobacco smoking, alcohol drinking, regular exercise, BMI (continuous), use of artificial denture, hypertension, diabetes, heart disease, cerebrovascular disease, respiratory disease, digestive system diseases, cancer, eye disease, arthritis and ADL disabled. Incidence rate (1000 person years).

**Figure 2 nutrients-14-01251-f002:**
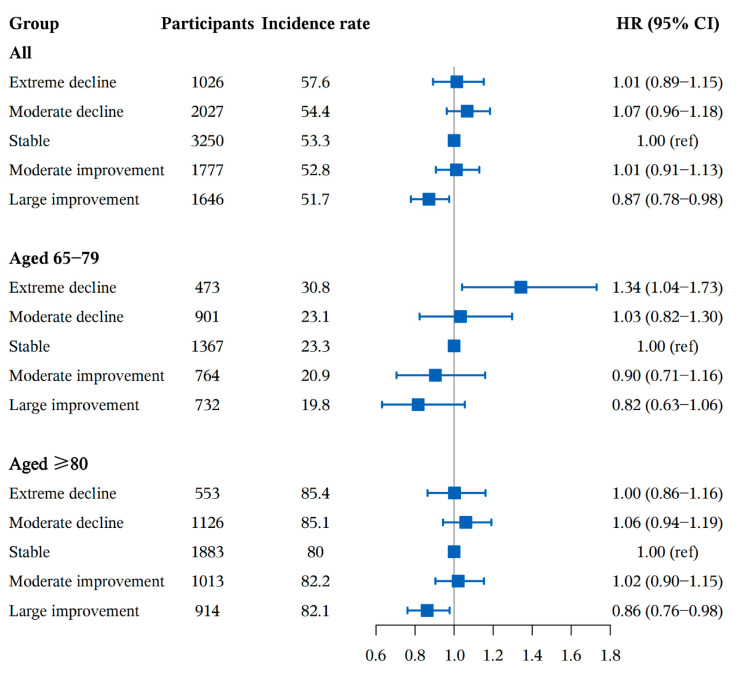
The association between absolute DDS change patterns and cognitive impairment. Adjusted for age (continuous), sex, residence, educational level, occupation, source of income, current marital status, living pattern, tobacco smoking, alcohol drinking, regular exercise, BMI (continuous), use of artificial denture, hypertension, diabetes, heart disease, cerebrovascular disease, respiratory disease, digestive system diseases, cancer, eye disease, arthritis and ADL disabled. Incidence rate (1000 person years).

**Table 1 nutrients-14-01251-t001:** Baseline characteristics of older people according to DDS change patterns (*n* = 9726).

Variables	Total			DDS Change Patterns from Baseline to First Follow Up						
		High-High	High-Medium	High-Low	Medium-High	Medium-Medium	Medium-Low	Low-High	Low-Medium	Low-Low
Number of participants	9726 (100.0)	1225 (12.6)	1175 (12.1)	115 (1.2)	1442 (14.8)	4012 (41.3)	743 (7.6)	139 (1.4)	616 (6.3)	259 (2.7)
Age in years, mean (SD)	80.0 (10.1)	76.2 (9.6)	78.6 (10.2)	80.2 (10.6)	78.6 (10.1)	81.2 (10.0)	82.3 (9.7)	80.1 (10.0)	82.2 (9.4)	83.9 (9.4)
Age group in years										
65–79	4237 (43.6)	767 (62.6)	601 (51.2)	52 (45.2)	719 (49.9)	1536 (38.3)	249 (33.5)	54 (38.9)	190 (30.8)	69 (26.6)
80–89	3614 (37.2)	318 (26.0)	371 (31.6)	39 (33.9)	487 (33.8)	1617 (40.3)	319 (42.9)	57 (41.0)	289 (46.9)	117 (45.2)
90–99	1449 (14.9)	116 (9.5)	157 (13.4)	17 (14.8)	179 (12.4)	658 (16.4)	132 (17.8)	23 (16.6)	113 (18.3)	54 (20.9)
≥100	426 (4.4)	24 (2.0)	46 (3.9)	7 (6.1)	57 (4.0)	201 (5.0)	43 (5.8)	5 (3.6)	24 (3.9)	19 (7.3)
Male	4644 (47.8)	748 (61.1)	625 (53.2)	38 (33.0)	772 (53.5)	1826 (45.5)	264 (35.5)	57 (41.0)	239 (38.8)	75 (29.0)
Urban Residence	2338 (24.0)	405 (33.1)	311 (26.5)	19 (16.5)	376 (26.1)	907 (22.6)	135 (18.2)	26 (18.7)	123 (20.0)	36 (13.9)
Education level										
No schooling	5328 (54.8)	435 (35.5)	562 (47.8)	69 (60.0)	685 (47.5)	2334 (58.2)	508 (68.4)	97 (69.8)	429 (69.6)	209 (80.7)
≤6 years	3316 (34.1)	512 (41.8)	450 (38.3)	34 (29.6)	539 (37.4)	1333 (33.2)	210 (28.3)	35 (25.2)	157 (25.5)	46 (17.8)
>6 years	1082 (11.1)	278 (22.7)	163 (13.9)	12 (10.4)	218 (15.1)	345 (8.6)	25 (3.4)	7 (5.0)	30 (4.9)	4 (1.5)
Occupation										
Worker	2707 (27.9)	273 (22.3)	252 (21.5)	19 (16.5)	343 (23.8)	1178 (29.4)	251 (33.8)	31 (22.3)	246 (39.9)	114 (44.0)
Farmer	4483 (46.1)	543 (44.3)	589 (50.2)	71 (61.7)	710 (49.3)	1838 (45.9)	331 (44.6)	86 (61.9)	223 (36.2)	92 (35.5)
Others	2529 (26.0)	409 (33.4)	333 (28.4)	25 (21.7)	388 (26.9)	991 (24.7)	161 (21.7)	22 (15.8)	147 (23.9)	53 (20.5)
Source of income										
Pension	2084 (21.4)	504 (41.1)	317 (27.0)	16 (13.9)	391 (27.1)	690 (17.2)	66 (8.9)	23 (16.6)	63 (10.2)	14 (5.4)
Other	7642 (78.6)	721 (58.9)	858 (73.0)	99 (86.1)	1051 (72.9)	3322 (82.8)	677 (91.1)	116 (83.5)	553 (89.8)	245 (94.6)
In marriage	4330 (44.5)	764 (62.4)	618 (52.6)	47 (40.9)	741 (51.4)	1583 (39.5)	234 (31.5)	65 (46.8)	203 (33.0)	75 (29.0)
Living pattern										
Living with family members	8166 (84.0)	1099 (89.7)	1040 (88.5)	98 (85.2)	1249 (86.6)	3334 (83.1)	569 (76.6)	110 (79.7)	477 (77.4)	190 (73.4)
Alone	1325 (13.6)	109 (8.9)	116 (9.9)	15 (13.0)	160 (11.1)	556 (13.9)	154 (20.7)	25 (18.1)	123 (20.0)	67 (25.9)
At nursing home	234 (2.4)	17 (1.4)	19 (1.6)	2 (1.7)	33 (2.3)	122 (3.0)	20 (2.7)	3 (2.2)	16 (2.6)	2 (0.8)
Tobacco smoking										
Current smoker	2295 (23.6)	343 (28.0)	305 (26.0)	24 (20.9)	373 (25.9)	899 (22.4)	136 (18.3)	31 (22.3)	135 (21.9)	49 (19.1)
Former smoker	1310 (13.5)	216 (17.6)	166 (14.1)	10 (8.7)	226 (15.7)	513 (12.8)	82 (11.0)	14 (10.1)	68 (11.0)	15 (5.8)
Nonsmoker	6114 (62.9)	666 (54.4)	703 (59.9)	81 (70.4)	842 (58.4)	2597 (64.8)	525 (70.7)	94 (67.6)	413 (67.1)	193 (75.1)
Alcohol drinking										
Current drinker	2391 (24.6)	400 (32.7)	336 (28.6)	24 (20.9)	368 (25.5)	915 (22.8)	158 (21.3)	31 (22.3)	124 (20.1)	35 (13.7)
Former drinker	870 (9.0)	106 (8.7)	96 (8.2)	7 (6.1)	157 (10.9)	364 (9.1)	60 (8.1)	12 (8.6)	52 (8.4)	16 (6.3)
Nondrinker	6454 (66.4)	719 (58.7)	743 (63.2)	84 (73.0)	916 (63.6)	2727 (68.1)	525 (70.7)	96 (69.1)	440 (71.4)	204 (80.0)
Regular exercises	3586 (36.9)	604 (49.3)	531 (45.2)	42 (36.5)	581 (40.3)	1400 (34.9)	200 (26.9)	32 (23.0)	144 (23.4)	52 (20.1)
Use of artificial denture	2951 (30.4)	478 (39.0)	401 (34.1)	29 (25.2)	510 (35.4)	1120 (27.9)	186 (25.1)	43 (30.9)	134 (21.8)	50 (19.3)
BMI, mean (SD), kg/m	21.8 (4.6)	22.7 (4.5)	22.3 (4.6)	21.5 (4.0)	21.7 (4.1)	21.7 (4.7)	21.8 (4.7)	20.9 (4.0)	21.2 (4.4)	21.4 (4.8)
Chronic diseases										
Hypertension	1643 (17.0)	246 (20.2)	198 (17.0)	13 (11.4)	261 (18.2)	632 (15.9)	115 (15.6)	24 (17.3)	110 (17.9)	44 (17.0)
Diabetes	193 (2.0)	37 (3.1)	27 (2.3)	1 (0.9)	41 (2.9)	65 (1.6)	8 (1.1)	3 (2.2)	9 (1.5)	2 (0.8)
Heart diseases	774 (8.0)	124 (10.2)	93 (8.0)	7 (6.1)	136 (9.5)	273 (6.9)	55 (7.5)	9 (6.5)	60 (9.8)	17 (6.6)
Cerebrovascular diseases	343 (3.6)	48 (4.0)	49 (4.2)	2 (1.8)	64 (4.5)	122 (3.1)	23 (3.1)	10 (7.2)	18 (2.9)	7 (2.7)
Respiratory diseases	991 (10.3)	131 (10.8)	107 (9.2)	13 (11.4)	163 (11.3)	397 (10.0)	78 (10.6)	10 (7.2)	70 (11.4)	22 (8.5)
Digestive system diseases	493 (5.1)	53 (4.4)	60 (5.2)	8 (7.0)	77 (5.4)	198 (5.0)	36 (4.9)	8 (5.8)	41 (6.7)	12 (4.7)
Cancer	23 (0.2)	7 (0.6)	3 (0.3)	0 (0.0)	3 (0.2)	8 (0.2)	0 (0.0)	0 (0.0)	2 (0.3)	0 (0.0)
ADL disabled	762 (7.8)	62 (5.1)	80 (6.8)	9 (7.8)	108 (7.5)	340 (8.5)	78 (10.5)	9 (6.5)	53 (8.6)	23 (8.9)
Eye diseases	1035 (10.7)	106 (8.7)	111 (9.5)	12 (10.5)	142 (9.9)	465 (11.7)	79 (10.7)	18 (13.0)	79 (12.9)	23 (8.9)
Arthritis	1708 (17.6)	218 (17.8)	201 (17.1)	27 (23.5)	245 (17.0)	695 (17.3)	139 (18.7)	24 (17.3)	114 (18.5)	45 (17.4)

Values are *n* (%) or mean (standard deviation, SD). DDS: dietary diversity score; BMI: body mass index; ADL: activities of daily living.

**Table 2 nutrients-14-01251-t002:** The association between DDS change patterns and cognitive impairment in nine foods.

Foods				DDS Change Patterns from Baseline to First Follow Up					
	Frequent-Frequent	Frequent-Occasional	Frequent-Rare	Occasional-Frequent	Occasional-Occasional	Occasional-Rare	Rare-Frequent	Rare-Occasional	Rare-Rare
Garlic									
No of cognitive impairment/person years	217/5865	223/4611	150/2846	254/5885	466/9033	436/6709	159/3225	365/6085	535/8064
Incidence rate	37.0	48.4	52.7	43.2	51.6	65.0	49.3	60.0	66.3
HR (95%CI)	1 (ref)	1.28 (1.06–1.55)	1.30 (1.05–1.60)	0.94 (0.78–1.13)	1.18 (1.00–1.40)	1.66 (1.41–1.96)	1.03 (0.83–1.26)	1.26 (1.06–1.50)	1.46 (1.24–1.72)
Fresh fruit									
No of cognitive impairment/person years	377/9201	246/5357	136/2651	318/6543	687/11117	366/5984	106/2186	306/4943	263/4343
Incidence rate	41.0	45.9	51.3	48.6	61.8	61.2	48.5	61.9	60.6
HR (95%CI)	1 (ref)	1.13 (0.96–1.33)	1.04 (0.85–1.27)	1.21 (1.04–1.42)	1.77 (1.54–2.02)	1.44 (1.24–1.68)	1.22 (0.98–1.52)	1.54 (1.31–1.82)	1.41 (1.19–1.68)
Tea									
No of cognitive impairment/person years	447/10003	132/2617	254/4899	168/3121	81/1839	242/3773	221/4501	215/3689	1045/17883
Incidence rate	44.7	50.4	51.9	53.8	44.0	64.1	49.1	58.3	58.4
HR (95%CI)	1 (ref)	1.12 (0.92–1.36)	0.99 (0.84–1.15)	1.00 (0.84–1.20)	0.86 (0.68–1.10)	1.21 (1.03–1.42)	0.86 (0.73–1.02)	1.21 (1.02–1.43)	0.96 (0.85–1.08)
Fresh vegetables									
No of cognitive impairment/person years	2089/41855	242/3778	71/894	249/3835	59/870	26/300	45/559	14/139	10/94
Incidence rate	49.9	64.1	79.4	64.9	67.8	86.6	80.5	100.5	106.0
HR (95%CI)	1 (ref)	1.20 (1.05–1.37)	1.03 (0.81–1.31)	1.15 (1.00–1.31)	1.37 (1.06–1.78)	1.62 (1.10–2.40)	1.09 (0.81–1.48)	2.43 (1.43–4.15)	2.65 (1.41–4.96)
Preserved vegetables									
No of cognitive impairment/person years	292/6751	237/4706	239/4441	237/4816	310/5804	368/6437	189/3933	284/5161	649/10278
Incidence rate	43.3	50.4	53.8	49.2	53.4	57.2	48.1	55.0	63.1
HR (95%CI)	1 (ref)	1.22 (1.03–1.45)	1.15 (0.97–1.36)	0.97 (0.81–1.15)	1.20 (1.02–1.41)	1.12 (0.96–1.31)	0.91 (0.75–1.09)	1.15 (0.97–1.36)	1.18 (1.02–1.36)
Beans									
No of cognitive impairment/person years	481/11407	358/7210	114/2214	496/9217	726/11303	240/3936	97/2050	182/3213	111/1775
Incidence rate	42.2	49.7	51.5	53.8	64.2	61.0	47.3	56.6	62.5
HR (95%CI)	1 (ref)	1.30 (1.13–1.49)	1.18 (0.96–1.45)	1.25 (1.10–1.42)	1.68 (1.48–1.90)	1.52 (1.29–1.79)	1.03 (0.82–1.29)	1.51 (1.26–1.82)	1.44 (1.16–1.79)
Fish									
No of cognitive impairment/person years	269/6540	215/4846	101/1720	275/5924	728/13423	391/6268	95/1973	346/5728	385/5904
Incidence rate	41.1	44.4	58.7	46.4	54.2	62.4	48.2	60.4	65.2
HR (95%CI)	1 (ref)	1.17 (0.98–1.40)	1.44 (1.14–1.81)	0.93 (0.78–1.11)	1.34 (1.15–1.55)	1.40 (1.19–1.64)	0.87 (0.68–1.10)	1.27 (1.07–1.50)	1.41 (1.19–1.68)
Meat									
No of cognitive impairment/person years	724/15434	306/5349	78/1587	434/8495	559/9567	245/3699	109/2301	176/3178	174/2714
Incidence rate	46.9	57.2	49.1	51.1	58.4	66.2	47.4	55.4	64.1
HR (95%CI)	1 (ref)	1.46 (1.27–1.67)	1.14 (0.90–1.44)	0.96 (0.85–1.08)	1.36 (1.21–1.53)	1.38 (1.19–1.61)	0.85 (0.69–1.04)	1.06 (0.89–1.27)	1.26 (1.06–1.51)
Eggs									
No of cognitive impairment/person years	758/15832	307/6074	78/1931	438/8396	568/9080	204/3285	125/2480	192/3112	135/2135
Incidence rate	47.9	50.5	40.4	52.2	62.6	62.1	50.4	61.7	63.2
HR (95%CI)	1 (ref)	1.11 (0.97–1.28)	0.91 (0.72–1.15)	0.98 (0.86–1.10)	1.43 (1.27–1.61)	1.39 (1.18–1.63)	1.03 (0.85–1.26)	1.45 (1.22–1.71)	1.32 (1.09–1.61)

DDS: dietary diversity score; ADL: activities of daily living; incidence rate (1000 person years). Adjusted for age (continuous), sex, residence, educational level, occupation, source of income, current marital status, living pattern, tobacco smoking, alcohol drinking, regular exercise, BMI (continuous), use of artificial denture, hypertension, diabetes, heart disease, cerebrovascular disease, respiratory disease, digestive system diseases, cancer, eye disease, arthritis and ADL disabled; the nine foods were mutually adjusted.

**Table 3 nutrients-14-01251-t003:** The association between DDS change patterns and cognitive impairment in subgroups.

Subgroups			DDS Change Patterns from Baseline to First Follow Up							*p* for Interaction
	High-High	High-Medium	High-Low	Medium-High	Medium-Medium	Medium-Low	Low-High	Low-Medium	Low-Low	
Age (years)										
65~79	1 (ref)	1.20 (0.89–1.63)	1.11 (0.55–2.24)	0.81 (0.58–1.11)	0.92 (0.70–1.21)	1.09 (0.74–1.62)	0.59 (0.21–1.62)	1.11 (0.72–1.71)	1.32 (0.73–2.39)	0.00
80~	1 (ref)	1.34 (1.09–1.64)	1.37 (0.91–2.05)	1.21 (1.00–1.48)	1.74 (1.47–2.07)	2.18 (1.76–2.69)	1.17 (0.80–1.71)	2.44 (1.96–3.04)	3.14 (2.43–4.07)	
Sex										
Male	1 (ref)	1.43 (1.13–1.82)	1.85 (0.99–3.45)	1.26 (0.99–1.60)	1.72 (1.40–2.12)	2.54 (1.92–3.37)	1.44 (0.82–2.51)	2.48 (1.84–3.33)	4.47 (3.06–6.51)	0.10
Female	1 (ref)	1.22 (0.96–1.55)	1.19 (0.78–1.83)	1.03 (0.82–1.30)	1.49 (1.22–1.82)	1.71 (1.34–2.18)	0.87 (0.55–1.38)	2.04 (1.59–2.62)	2.23 (1.66–3.00)	
Marital status										
Married	1 (ref)	1.07 (0.82–1.41)	1.12 (0.56–2.24)	0.93 (0.71–1.23)	1.12 (0.88–1.42)	1.13 (0.80–1.61)	0.73 (0.37–1.41)	1.48 (1.04–2.10)	2.14 (1.37–3.34)	0.75
Not married	1 (ref)	1.02 (0.82–1.27)	0.84 (0.56–1.27)	0.85 (0.68–1.05)	0.94 (0.78–1.14)	1.02 (0.81–1.28)	0.96 (0.63–1.44)	1.12 (0.88–1.42)	1.09 (0.82–1.44)	
Tobacco smoking										
Current or former smoker	1 (ref)	1.02 (0.77–1.35)	0.68 (0.36–1.28)	0.80 (0.61–1.05)	0.97 (0.77–1.23)	1.15 (0.83–1.58)	1.13 (0.61–2.07)	1.27 (0.91–1.77)	1.93 (1.24–3.02)	0.29
Non-smoker	1 (ref)	1.06 (0.85–1.31)	1.07 (0.70–1.63)	0.93 (0.75–1.15)	1.04 (0.86–1.25)	1.13 (0.90–1.42)	0.92 (0.60–1.41)	1.28 (1.01–1.63)	1.28 (0.96–1.70)	
Alcohol drinking										
Current or former drinker	1 (ref)	0.95 (0.72–1.25)	0.82 (0.44–1.55)	0.79 (0.60–1.05)	0.89 (0.70–1.13)	0.88 (0.64–1.23)	0.79 (0.41–1.52)	0.99 (0.71–1.39)	1.48 (0.96–2.29)	0.45
Non-drinker	1 (ref)	1.08 (0.87–1.34)	1.04 (0.68–1.59)	0.93 (0.75–1.15)	1.10 (0.91–1.32)	1.25 (0.99–1.57)	1.01 (0.67–1.53)	1.48 (1.16–1.88)	1.35 (1.02–1.80)	
Regular exercises										
Yes	1 (ref)	1.00 (0.77–1.29)	0.79 (0.46–1.35)	0.87 (0.67–1.12)	1.14 (0.91–1.43)	1.39 (1.01–1.90)	1.15 (0.56–2.38)	1.60 (1.12–2.29)	2.04 (1.26–3.30)	0.00
No	1 (ref)	1.01 (0.81–1.27)	1.05 (0.66–1.68)	0.83 (0.67–1.03)	0.90 (0.74–1.09)	0.96 (0.76–1.21)	0.83 (0.56–1.24)	1.12 (0.88–1.42)	1.18 (0.89–1.56)	
Use of artificial denture										
Yes	1 (ref)	1.05 (0.79–1.39)	1.60 (0.82–3.12)	1.02 (0.78–1.34)	1.17 (0.92–1.49)	1.24 (0.86–1.79)	1.29 (0.64–2.57)	1.54 (1.03–2.29)	1.03 (0.55–1.92)	0.92
No	1 (ref)	1.02 (0.82–1.27)	0.78 (0.51–1.18)	0.81 (0.66–1.01)	0.95 (0.79–1.14)	1.04 (0.83–1.30)	0.85 (0.57–1.28)	1.19 (0.94–1.50)	1.37 (1.05–1.79)	
ADL disabled										
Yes	1 (ref)	0.80 (0.48–1.31)	1.64 (0.69–3.90)	0.71 (0.44–1.15)	0.72 (0.47–1.09)	0.95 (0.57–1.58)	0.34 (0.10–1.16)	1.11 (0.66–1.89)	1.44 (0.75–2.77)	0.04
No	1 (ref)	1.08 (0.90–1.30)	0.84 (0.57–1.24)	0.90 (0.75–1.08)	1.07 (0.91–1.25)	1.09 (0.89–1.34)	0.98 (0.68–1.41)	1.26 (1.03–1.56)	1.30 (1.01–1.68)	

DDS: dietary diversity score; ADL: activities of daily living; BMI: body mass index. Adjusted for age (continuous), sex, residence, educational level, occupation, source of income, current marital status, living pattern, tobacco smoking, alcohol drinking, regular exercise, BMI (continuous), use of artificial denture, hypertension, diabetes, heart disease, cerebrovascular disease, respiratory disease, digestive system diseases, cancer, eye disease, arthritis and ADL disabled.

## Data Availability

Available from the Peking University on request (https://opendata.pku.edu.cn/).

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
