# Peer review of "Association between Dietary Diversity Changes and Cognitive Impairment among Older People: Findings from a Nationwide Cohort Study"

_nutrients, 2022, doi:10.3390/nu14061251_

Round 1

Reviewer 1 Report

1. Please discuss the following papers in the background and discussion:

https://www.sciencedirect.com/science/article/pii/S155252601833560X

https://academic.oup.com/ajcn/article/114/3/871/6277979

https://www.ncbi.nlm.nih.gov/pmc/articles/PMC6179018/

https://journals.sagepub.com/doi/full/10.1177/1533317516673155

2. Please address whether dietary diversity score is associated with total caloric intake. I am wondering whether total daily caloric intake is associated with cognitive impairment. Please provide data if possible. Otherwise, please discuss the issue in detail.

Author Response

Response to Reviewer 1 Comments

Point 1: Please discuss the following papers in the background and discussion:

https://www.sciencedirect.com/science/article/pii/S155252601833560X

https://academic.oup.com/ajcn/article/114/3/871/6277979

https://www.ncbi.nlm.nih.gov/pmc/articles/PMC6179018//

https://journals.sagepub.com/doi/full/10.1177/1533317516673155

Response 1: We are grateful for the suggestion, and we have discussed the above papers in the background and discussion as follows “Accumulated evidence supports an association between high-quality diet and cognitive impairment, and indicates that higher adherence to the high-quality diet to be associated with better cognition in older people [29-31].”(Lines 73-76, Page 2). “However, a review shows that dietary changes are present in most people with dementia, with the progression of dementia, the patient’s ability to obtain adequate nutrition decreases. This could potentially result in reverse causation bias, where changes in dietary may be a consequence of cognitive impairment rather than a cause. Therefore, we exclude those who developed cognitive impairment within the fourth year of follow-up to evaluate whether the relationships are consistent over time, and the sensitivity analyses were robust [42].” (Lines 49-57, Page 19). “Overall, improving DDS or maintaining high DDS is critical for adequate nutritional intake, and thus might reduce the risk of cognitive impairment, this is in accordance with the findings of a multidomain intervention trial in Finnish [31].” (Lines 57-61, Page 19).

Reference:

[29]     Dominguez, L.J.; Barbagallo, M. Nutritional prevention of cognitive decline and dementia. Acta Biomed 2018, 89, 276-290, DOI:10.23750/abm.v89i2.7401.

[30]     Nooyens, A.C.J.; Yildiz, B.; Hendriks, L.G.; Bas, S.; van Boxtel, M.P.J.; Picavet, H.S.J.; Boer, J.M.A.; Verschuren, W.M.M. Adherence to dietary guidelines and cognitive decline from middle age: the Doetinchem Cohort Study. Am J Clin Nutr 2021, 114, 871-881, DOI:10.1093/ajcn/nqab109.

[31]     Lehtisalo, J.; Levälahti, E.; Lindström, J.; Hänninen, T.; Paajanen, T.; Peltonen, M.; Antikainen, R.; Laatikainen, T.; Strandberg, T.; Soininen, H., et al. Dietary changes and cognition over 2 years within a multidomain intervention trial—The Finnish Geriatric Intervention Study to Prevent Cognitive Impairment and Disability (FINGER). Alzheimer's & Dementia 2019, 15, 410-417, DOI:10.1016/j.jalz.2018.10.001.

[42]     Cipriani, G.; Carlesi, C.; Lucetti, C.; Danti, S.; Nuti, A. Eating Behaviors and Dietary Changes in Patients With Dementia. Am J Alzheimers Dis Other Demen 2016, 31, 706-716, DOI:10.1177/1533317516673155.

Point 2: Please address whether dietary diversity score is associated with total caloric intake. I am wondering whether total daily caloric intake is associated with cognitive impairment. Please provide data if possible. Otherwise, please discuss the issue in detail.

Response 2: Thank you for the valuable comment. Some studies had shown that high caloric intake is associated with an increased risk of cognitive impairment [1-3]. However, given that the means age of the participants was as high as 80.0 years and the low education level, we considered that semiquantitative FFQs measured might be difficult for the elderly, and DDS without quantitative measurements might be the most sensible choice. Therefore, the collected dietary information in our study lacks quantitative dietary intake, and we couldn’t address whether DDS is associated with total caloric intake. We have added the discussion in limitations as follows “the collected dietary information lacks quantitative dietary intake, which was not available from the FFQ, thus we were unable to adjust the energy intake model and address whether DDS is associated with total caloric intake. However, a number of key determinants of energy intake were taken into account, such as age, sex, BMI, comorbidities and physical activity [44,49].” (Lines 110-113, Page 20).

Reference:

[1]       Roberts, R.O.; Roberts, L.A.; Geda, Y.E.; Cha, R.H.; Pankratz, V.S.; O'Connor, H.M.; Knopman, D.S.; Petersen, R.C. Relative intake of macronutrients impacts risk of mild cognitive impairment or dementia. J Alzheimers Dis 2012, 32, 329-339, DOI:10.3233/jad-2012-120862.

[2]       Ding, B.; Xiao, R.; Ma, W.; Zhao, L.; Bi, Y.; Zhang, Y. The association between macronutrient intake and cognition in individuals aged under 65 in China: a cross-sectional study. BMJ Open 2018, 8, e018573, DOI:10.1136/bmjopen-2017-018573.

[3]       Yeh, T.S.; Yuan, C.; Ascherio, A.; Rosner, B.A.; Blacker, D.; Willett, W.C. Long-term intake of total energy and fat in relation to subjective cognitive decline. Eur J Epidemiol 2021, 10.1007/s10654-021-00814-9, DOI:10.1007/s10654-021-00814-9.

[44]     Rhee, J.J.; Cho, E.; Willett, W.C. Energy adjustment of nutrient intakes is preferable to adjustment using body weight and physical activity in epidemiological analyses. Public Health Nutr 2014, 17, 1054-1060, DOI:10.1017/s1368980013001390.

[49]     Jakes, R.W.; Day, N.E.; Luben, R.; Welch, A.; Bingham, S.; Mitchell, J.; Hennings, S.; Rennie, K.; Wareham, N.J. Adjusting for energy intake--what measure to use in nutritional epidemiological studies? Int J Epidemiol 2004, 33, 1382-1386, DOI:10.1093/ije/dyh181.

Reviewer 2 Report

The text requires proofreading, there are mistakes, missing words (e.g. in the first sentence of abstract).

Could you specify what a "person-years" concept is? I did not grasp it from the current description of methodology and analysis.

Author Response

Response to Reviewer 2 Comments

Point 1: The text requires proofreading, there are mistakes, missing words (e.g. in the first sentence of abstract).

Response 1: Thank you for your comment. We had checked the full text and all typos and missing words have been revised.

Point 2: Could you specify what a "person-years" concept is? I did not grasp it from the current description of methodology and analysis.

Response 2: Thank you for the valuable comment. "Person-years" are calculated from the time of the baseline survey of participants to the earliest of the following events (first occurrence of cognitive impairment, death, lost to follow-up or time of the last survey). And the specific calculation is as follows, Person-years =(year of the events [first occurrence of cognitive impairment, death, lost to follow-up or time of the last survey]− year of baseline survey) + (month of the events [first occurrence of cognitive impairment, death, lost to follow-up or time of the last survey]− month of baseline survey)/12 + (day of the events [first occurrence of cognitive impairment, death, lost to follow-up or time of the last survey]− day of baseline survey)/365.

We have added the description of "Person-years" in statistical analysis section.
